# Steroidogenic Factor 1 (NR5A1) Activates ATF3 Transcriptional Activity

**DOI:** 10.3390/ijms21041429

**Published:** 2020-02-20

**Authors:** Natsuko Emura, Chiung-Min Wang, William Harry Yang, Wei-Hsiung Yang

**Affiliations:** 1The United Graduate School of Agricultural Sciences, Iwate University, Morioka, Iwate 020-8550, Japan; u3118002@iwate-u.ac.jp; 2Department of Biomedical Sciences, Mercer University School of Medicine, Savannah, GA 31404, USA; meowy200@yahoo.com (C.-M.W.); meowy100@yahoo.com (W.H.Y.)

**Keywords:** NR5A1, ATF3, interaction, transcriptional activity, SUMOylation, phosphorylation

## Abstract

Steroidogenic Factor 1 (SF-1/NR5A1), an orphan nuclear receptor, is important for sexual differentiation and the development of multiple endocrine organs, as well as cell proliferation in cancer cells. Activating transcription factor 3 (ATF3) is a transcriptional repressor, and its expression is rapidly induced by DNA damage and oncogenic stimuli. Since both NR5A1 and ATF3 can regulate and cooperate with several transcription factors, we hypothesized that NR5A1 may interact with ATF3 and plays a functional role in cancer development. First, we found that NR5A1 physically interacts with ATF3. We further demonstrated that ATF3 expression is up-regulated by NR5A1. Moreover, the promoter activity of the *ATF3* is activated by NR5A1 in a dose-dependent manner in several cell lines. By mapping the *ATF3* promoter as well as the site-directed mutagenesis analysis, we provide evidence that NR5A1 response elements (−695 bp and −665 bp) are required for *ATF3* expression by NR5A1. It is well known that the transcriptional activities of NR5A1 are modulated by post-translational modifications, such as small ubiquitin-related modifier (SUMO) modification and phosphorylation. Notably, we found that both SUMOylation and phosphorylation of NR5A1 play roles, at least in part, for NR5A1-mediated *ATF3* expression. Overall, our results provide the first evidence of a novel relationship between NR5A1 and ATF3.

## 1. Introduction

Steroidogenic factor 1 (NR5A1) (also called SF-1) is an orphan nuclear receptor superfamily critical for regulation of sex determination, adrenal and gonadal development, reproductive function, and steroidogenesis [1,2,3,4]. Furthermore, recent studies have indicated that NR5A1 is associated with adrenocortical cancer development [5,6,7,8]. NR5A1 targets a variety of genes such as CYP17, DAX-1, and STAR [9,10,11,12,13], and regulates transcription of these genes through cooperating with multiple transcription factors and cofactors, including GATA4, SOX9, EGR1, and WT1 [14,15,16,17]; however, the molecular mechanisms of NR5A1 in cancer development are still largely unknown.

The majority of the transcription factors are regulated by post-translational modifications which are essential for normal physiological functions in cells. Small ubiquitin-related modifier (SUMO) modification is one type of post-translational modifications, and SUMOylation of transcription factors and nuclear receptors has a strong impact on their transcriptional activities [18,19,20]. Mounting evidence has demonstrated that NR5A1 can be SUMOylated on lysine 194 (K194, the major site) and lysine 119 (K119, the minor site), and SUMO modification of NR5A1 regulates its transcriptional activities [21,22,23,24]. In addition to the SUMOylation, phosphorylation at serine 203 (S203) plays a key in the transcriptional capacity of NR5A1 [25,26]. Moreover, several reports have demonstrated that NR5A1 activity through phosphorylation at S203 is inhibited by SUMOylation of NR5A1 [27,28], suggesting the important interaction between SUMOylation and phosphorylation. These evidence indicates that post-translational modifications such as phosphorylation and SUMOylation of NR5A1 may affect NR5A1 function in cancer cells.

Activating transcription factor 3 (ATF3), which is a member of the ATF/CREB family of transcriptional repressors, binds the ATF/cAMP response element (CRE) of a number of promoters to regulate its downstream target genes [29]. ATF3 is rapidly induced in cells once exposed to stress stimuli including those initiated by cytokines, genotoxic agents, infections, nerve injury, tissue damage, or physiological stresses [29,30]. In addition, a solid line of evidence has indicated that ATF3 is able to suppress cell growth and inhibit the development of tumors [31,32,33]; however, some evidence has implicated that ATF3 is up-regulated in many cancers, suggesting that ATF3 is an oncogene [34,35]. These results imply that ATF3 expression has positive and negative effects on proliferation and survival of cancer cells.

In the present work, we report the relationship between NR5A1 and ATF3. NR5A1 physically interacts with ATF3, and NR5A1 binds *ATF3* promoter to enhance the *ATF3* transcriptional activity. Interestingly, we have demonstrated that both phosphorylation and SUMOylation of NR5A1 have functions in NR5A1-mediated *ATF3* expression. Taken together, these results support a novel relationship between NR5A1 and ATF3 and this relationship may have an impact in organ differentiation and cancer development. 

## 2. Results

### 2.1. Steroidogenic Factor 1 (NR5A1) Physically Interacts with Activating Transcription Factor 3 (ATF3)

Since NR5A1 cooperates with several transcription factors [14,15,36,37], we first analyzed whether NR5A1 can bind to ATF3. To address this question, we co-transfected HIS-FLAG-tagged *ATF3* with or without HA-tagged *NR5A1* expression plasmids into H1299 cells, which do not express endogenous NR5A1, and Ni^2+^-bead pull-down assay was performed. As shown in Figure 1, when both ATF3 and NR5A1 were expressed, the NR5A1 band was observed in Ni^2+^-bead pull-down samples. To further investigate whether the ATF3 physically binds to NR5A1, HIS-FLAG-tagged *ATF3* was expressed in the Y1 cells, which endogenously express NR5A1, and Ni^2+^-bead pull-down assay was performed. As shown in Figure 1, HIS-FLAG-tagged ATF3 was expressed in Y1 cells and was precipitated (WCL and Ni^2+^-bead pull-down). NR5A1 was also present in the cell lysate (WCL) and co-precipitated with HIS-FLAG-tagged ATF3 (Ni^2+^-bead pull-down). Notably, the endogenous NR5A1 on Y1 cells is smaller than HA-tagged NR5A1 on H1299 cells. These results indicate that NR5A1 physically binds to ATF3.

### 2.2. NR5A1 Increases ATF3 Protein Level

Since NR5A1 is a transcription factor and binds to ATF3, we next investigated the role of NR5A1 on ATF3 expression. Expression plasmids encoding wild-type *NR5A1* or empty plasmids were transfected into H1299, MCF7, and Y1 cells. As shown in Figure 2, expression of wild-type *NR5A1* led to increase ATF3 protein levels in H1299, MCF7, and Y1 cells (approximately 1.8–2.0 fold). This finding indicates that NR5A1 has potential to promote ATF3 expression.

### 2.3. NR5A1 Is an Activator of the ATF3 Promoter

As NR5A1 increases ATF3 expression, we next investigated the role of NR5A1 on *ATF3* promoter activation. The −1372 bp *ATF3* promoter-LUC reporter plasmid was co-transfected with *NR5A1* expression plasmid into several different cell lines and *ATF3* promoter activity was determined by measuring the LUC activity in cell lysates 48 h after transfection. As shown in Figure 3A–C, expression of NR5A1 generated a dose-dependent increase in the activity of *ATF3* gene transcription. This finding indicates that NR5A1 is an activator of the *ATF3* transcription.

### 2.4. Minimal ATF3 Promoter Region Responsive to NR5A1 Activation

Because the −1372 bp *ATF3* promoter contains two potential candidates of NR5A1 response elements (REs), the *ATF3* promoter was truncated to determine the minimal region that is important for transcriptional activation by NR5A1 (Figure 4A). Deletion of the distal NR5A1 RE (−695 bp), as shown in the −680 bp promoter, resulted in a slight loss of NR5A1-mediated *ATF3* transcriptional activity. However, when both NR5A1 REs were truncated, as shown in the −300 bp promoter, the NR5A1-mediated *ATF3* transcriptional activity was significantly decreased (approximately 75% loss). 

To further determine whether the two NR5A1 REs are required for NR5A1-mediated *ATF3* expression, we first searching potential NR5A1 binding site(s) on *ATF3* promoter region using ALGGEN-PROMO website (http://alggen.lsi.upc.es/cgi-bin/promo_v3/promo/promoinit.cgi?dirDB=TF_8.3). We identified two potential NR5A1 binding sites on the human *ATF3* promoter region. The two potential NR5A1 binding sites are located 695 bp (CGCCGCAGAGGTCACACCCGG) and 665 bp (TGACTTTGGACACCTTCCCC) upstream of the transcriptional start site, suggesting that NR5A1 may regulate *ATF3* transcription directly. We next generated −695 bp mutant (CAGAGGTCA→CAGAAATCA) and −665 bp mutant (GGACACCTTC→GGACAAATTC) *ATF3* promoter-LUC reporter plasmids. As shown in Figure 4B, mutation of either −695 bp or −665 bp NR5A1 RE resulted in approximately 40% of NR5A1-mediated *ATF3* promoter activity. Notably, mutations of both −695 bp and −665 bp NR5A1 REs dramatically reduced *ATF3* promoter activity (approximately 70–75% loss). Together, these results indicate that both −695 bp and −665 bp REs are essential for the NR5A1 action on the *ATF3* promoter.

### 2.5. Phosphorylation and Small Ubiquitin-Related Modifier (SUMO)ylation of NR5A1 Are Required for Full NR5A1-Mediated ATF3 Transcriptional Activity

Because NR5A1 has been demonstrated to be modified by post-translational modifications such as phosphorylation and SUMOylation [21,22,23,24,25,26], we next examined the effect of the post-translational modifications of NR5A1 on its transcriptional activity of the *ATF3* promoter. H1299 and Saos2 cells were co-transfected the *ATF3* promoter-LUC reporter plasmid with either wild-type (WT), K119R (mimicking de-SUMOylated at K119), K194R (mimicking de-SUMOylated at K194), 2KR (mimicking de-SUMOylated at both K119 and K194), S203A (mimicking de-phosphorylated at S203), or SUMO-WT (mimicking SUMOylated) *NR5A1* expression plasmid. As shown in Figure 5A and 5B, in both H1299 and Saos2 cell lines, while the WT and SUMO-WT NR5A1 enhanced *ATF3* promoter activity, K119R, K194R, 2KR, and S203A NR5A1 reduced this effect. Interestingly, loss of phosphorylation reduced more NR5A1-mediated *ATF3* promoter activity than loss of SUMOylation did, suggesting that phosphorylation is more important for NR5A1 on *ATF3* promoter activity. These results suggest that both phosphorylation and SUMOylation are essential, at least in part, for NR5A1-mediated *ATF3* promoter activity.

## 3. Discussion

The growth and development of tissue and organisms requires precisely orchestrated gene regulation and cell proliferation. NR5A1, an orphan member of the nuclear receptor superfamily, is expressed in the steroidogenic adrenals and gonads specifically [4,38,39]. Although NR5A1 has numerous functions including sex determination, adrenal development, and steroidogenesis [1,2,3,4], it is also involved in cancer development [5,6,7,8]. Mutations of *NR5A1* lead to many disorders and phenotypes, including premature ovarian failure, adrenocortical insufficiency, sex reversal, and spermatogenesis failure. Herein, we demonstrate a novel NR5A1 and ATF3 interaction, and post-translational modifications such as phosphorylation and SUMOylation are involved in this relationship between NR5A1 and ATF3.

NR5A1 can cooperate with several transcription factors [14,15,16,17], and in the present study, we observed the physical interaction of NR5A1 with ATF3. Another AP-1 protein family member, JDP2, also binds to NR5A1 (data not shown), suggesting that the members of AP-1 protein family may have potential binding to NR5A1. NR5A1 is overexpressed in adrenocortical tumors [5,6,7], suggesting that NR5A1 relates to cancer development. Moreover, ATF3 is induced by stress stimuli and associates with various kinds of cancer development [29,30,34]. Previous reports have demonstrated that ATF3 is expressed in H295R human adrenal carcinoma cells [40] and may be involved in adrenocortical aldosterone synthesis [41]. In addition, co-transfection of ATF3 and a member of the AP-1 complex JUNB into H295R cells increases *STAR* expression, which is also a downstream target of NR5A1 [13,40]. Taken together, NR5A1-ATF3 interaction and axis may regulate cell population in adrenocortical cancer cells through regulation of cell proliferation related genes. However, whether NR5A1-ATF3 interaction regulates NR5A1-mediated ATF3 expression in a feedback loop remains unknown. Further future studies are indeed necessary to dissect whether NR5A1-ATF3 axis directly regulates aldosterone synthesis in adrenals and is directly involved in cancer development.

It is well known that NR5A1, as a critical master regulator, helps control the activities of numerous genes related to the development of the adrenals and the ovaries (testes and ovaries) [9,10,11,12,13]. In the present work, we showed that NR5A1 up-regulates ATF3 protein level and enhances *ATF3* promoter activity. Our promoter analysis further supports that both NR5A1 response elements −695 bp and −665 bp on the *ATF3* promoter region are essential for regulating *ATF3* gene expression. Our data highlights that *ATF3* is the novel target gene for NR5A1.

Post-translational modifications such as phosphorylation, SUMOylation, and acetylation influence wide range of cellular activities, including cancer development [42,43]. It has been determined that NR5A1 can be SUMOylated on K194 and K119, which are major and minor sites, respectively, and the SUMOylation of NR5A1 influences its transcriptional activities [21,22,23,24]. In the present work, we demonstrated that replacement of K194 and/or K119 by an arginine residue in NR5A1 leads to reduce *ATF3′*s transcriptional activity, suggesting that SUMOylation of NR5A1 plays an important role for *ATF3* expression. Moreover, as a transcription factor, NR5A1 has been shown to be phosphorylated at S203 [25,26]. Importantly, loss of phosphorylation on NR5A1 S203 (S203A) significantly decreases its activity in regulating the *ATF3* promoter more than SUMOylation mutants. This result, consistent with previous reports, further highlights the crucial role of phosphorylation on NR5A1 function as a transcription factor. The acetylation of NR5A1 has also been studied extensively. The major acetylation sites on NR5A1 based on reports are K34, K38, and K72 [44] and amino acids from 56–63 and 102–106 [45]. Overall, we provide the first evidence that phosphorylation and SUMOylation of NR5A1 play an important function in regulation of *ATF3* gene activity.

In conclusion, this investigation demonstrated that NR5A1 physically interacts with ATF3. Our study also identified that NR5A1 is a novel activator of *ATF3* promoter, and that the post-translational modifications (phosphorylation and SUMOylation) play a critical role for NR5A1′s transcriptional activity.

## 4. Materials and Methods 

### 4.1. Chemicals and Reagents

Cell culture media and reagents were purchased from Thermo Fisher Scientific (Waltham, MA, USA). Antibodies against ATF3 and β-Actin (Santa Cruz Biotechnology Inc., Santa Cruz, CA, USA), HIS (Origene, Rockville, MD, USA) and NR5A1 (Upstate Biochemistry Inc., Charlottsville, VA, USA) were purchased commercially. Luciferase activity was measured using the Dual-Luciferase Reporter Assay System (Promega, Madison, WI, USA).

### 4.2. DNA Constructs

HIS-FLAG *NR5A1*-pcDNA3 expression plasmid was generated by PCR cloning and described previously [21]. S203A, K119R, K194R, and 2KR *NR5A1* expression plasmids were created by PCR-based mutagenesis (QuikChange Lightning site-directed mutagenesis kit, Agilent/Strategene, La Jolla, CA, USA). *SUMO*-WT *NR5A1* expression plasmid was created by inserting *SUMO1* DNA sequence into the N-terminal *NR5A1* DNA sequence. The human *ATF3* promoter (−1372/+22 bp) pGL2 plasmid was kindly provided by Dr. Aronheim (Technion-Israel Institute of Technology, Haifa, Israel). The human *ATF3* promoter deletion constructs were then generated by removal of specific fragments of DNA sequence in Yang lab. The human *ATF3* promoter with NR5A1 RE mutant plasmids were created by PCR-based mutagenesis (QuikChange Lightning site-directed mutagenesis kit, Agilent/Strategene, La Jolla, CA, USA). All constructs were verified by Sanger nucleotide sequencing.

### 4.3. Cell Culture and Transfection

H1299, HepG2, MCF7, Y1, and Saos2 cells were obtained from the American Type Culture Collection (Manassas, VA, USA). The cells were maintained in Dulbecco’s-modified Eagle medium (DMEM) in the presence of 10% fetal bovine serum and Pen/Strep antibiotics (GIBCO/Life Technologies, Grand Island, NY, USA) in humidified air containing 5% CO_2_ at 37 °C and cultured for less than six months. After incubation, the cells were transfected with specific expression plasmids described in each assay using Fugene HD Transfection Reagent (Roche, Madison, WI, USA). Forty-eight hours after transfection, the cells were harvested and lysed.

### 4.4. ATF3 Promoter luciferase Reporter Assays

Cells were cultured in 24-well plates overnight and then transiently transfected with *ATF3* promoter-firefly luciferase plasmid and internal control pRL-TK plasmid (which encodes *Renilla* luciferase activity) in the presence of Fugene HD Transfection Reagent (Roche, Madison, WI, USA). At 48 h after transfection, the cells were harvested and lysed in passive lysis buffer (Promega). Luminescence was detected with the Dual-Luciferase Reporter Assay System (Promega) according to the manufacturer’s instructions. Firefly luciferase activity was normalized by calculating the ratio to *Renilla* luciferase activity. The relative luciferase activity was calculated as a fold change to the control groups. All experiments were performed three times in triplicate setting.

### 4.5. Immunoprecipitation Assays

Y1, MCF7, and H1299 cells (2 × 10^6^) were seeded onto 100 mm plates. Forty-eight hours after transfection, cells were harvested and lysed in Sabatini lysis buffer (40 mM HEPES, 120 mM sodium chloride, 10 mM sodium glycerophosphate, 10 mM sodium pyrophosphate, 50 mM sodium fluoride, 0.5 mM sodium orthovanadate, 1 mM EDTA, 1% Triton X-100) containing protease inhibitor cocktail (Sigma, St. Louis, MO, USA), followed by rotation for 1 h at 4 °C to solubilize proteins. Soluble proteins were collected and immunoprecipitated with the indicated antibody overnight. Protein A agarose beads were added to protein lysates for 2 h at 4 °C. Beads were centrifuged and washed at least three times with lysis buffer. For Ni^2+^-bead pull-down assays, Ni^2+^-NTA agarose was used to precipitate HIS-tagged ATF3 or HIS-tagged NR5A1 from cell lysates. Proteins were eluted by boiling in 40 µL of 2 × Laemmli sample buffer, resolved by 8–10% sodium dodecyl sulfate-polyacrylamide gel electrophoresis (SDS-PAGE), and processed for immunoblotting as described below.

### 4.6. Western Blot Analysis

Protein lysates were allowed to rotate at 4 °C for 1 h, and protein contents of the high-speed supernatant were determined using the BCA^TM^ Protein Assay kit assay (Pierced/Thermo Scientific, Rockford, IL, USA). Equivalent quantities of protein (40 µg) were resolved on polyacrylamide-SDS gels, transferred to polyvinylidene difluoride (PVDF) membrane (Bio-Rad, Hercules, CA, USA), and immunoblotted with specific antibodies. Results were visualized using the Supersignal West Dura Extended Duration Substrate kit (Pierce Chemical Co., Rockford, IL, USA). The intensity of the protein band was quantified by ImageJ program.

### 4.7. Statistical Analysis

We performed statistical analysis for the significance of the differences between two measurements by using the Student’s *t* test. *p* < 0.05 was considered statistically significant between groups.

## 5. Conclusions 

In summary, we have shown a novel relationship between NR5A1 and ATF3 for the first time. Our studies suggest the possibility of interaction between NR5A1 and ATF3 associated with cell proliferation in adrenocortical cancer cells. Moreover, NR5A1 is a novel activator of the *ATF3* promoter, and the NR5A1-mediated *ATF3* transcription is, at least in part, regulated by the post-translational modifications (phosphorylation and SUMOylation). Together, our findings add a new layer of information to the previous understanding of how NR5A1 functions in steroidogenesis, reproductive organ differentiation, and even cancer development.

## Figures and Tables

**Figure 1 ijms-21-01429-f001:**
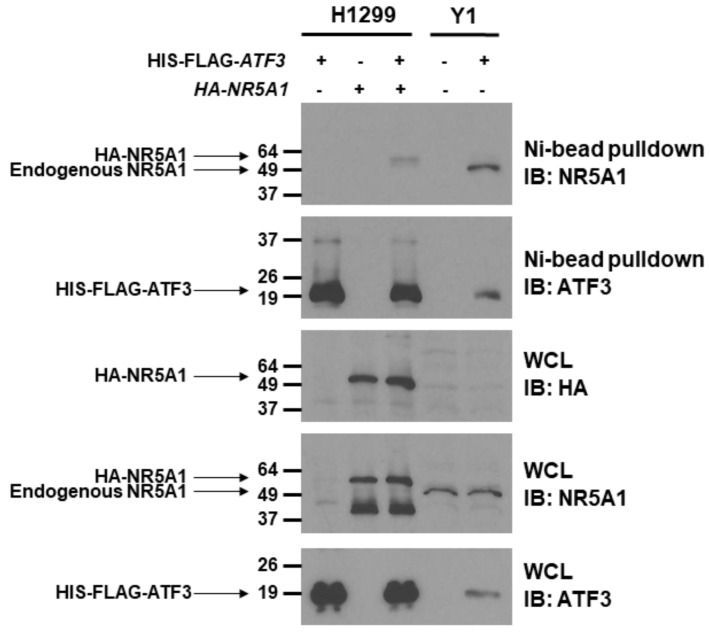
Steroidogenic factor 1 (NR5A1) physically interacts with activating transcription factor 3 (ATF3). H1299 cells were transfected with of HIS-FLAG-tagged *ATF3* or HA-tagged *NR5A1* or both expression plasmids. Y1 cells were transfected with HIS-FLAG-tagged *ATF3* expression plasmid. Forty-eight hours later, the cell lysates were immunoprecipitated by anti-HIS antibody, followed by anti-ATF3 or anti-NR5A1 immunoblotting. WCL indicates whole cell lysates.

**Figure 2 ijms-21-01429-f002:**
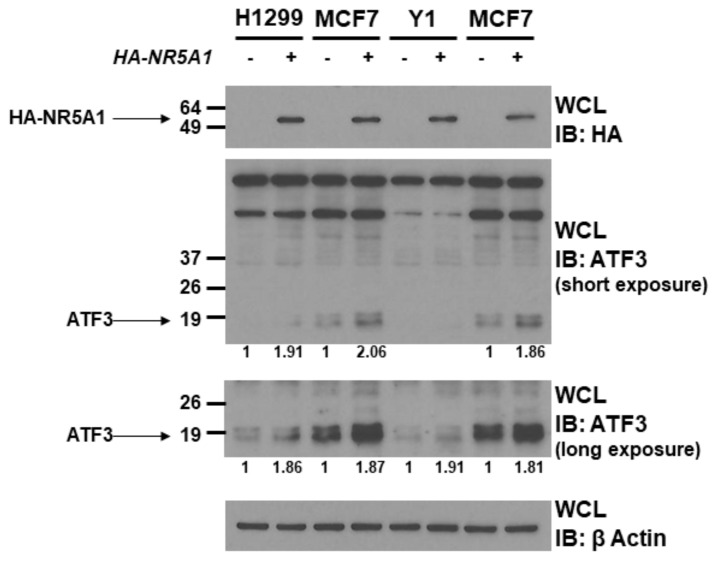
NR5A1 increases ATF3 protein level. H1299, MCF7, and Y1 cells were transfected with either pcDNA3 vector or wild-type HA-tagged *NR5A1* expression plasmid. Forty-eight hours later, the expression levels of NR5A1 and ATF3 were determined using anti-HA and anti-ATF3 immunoblotting, respectively. The β-Actin levels were also determined for equal loading.

**Figure 3 ijms-21-01429-f003:**
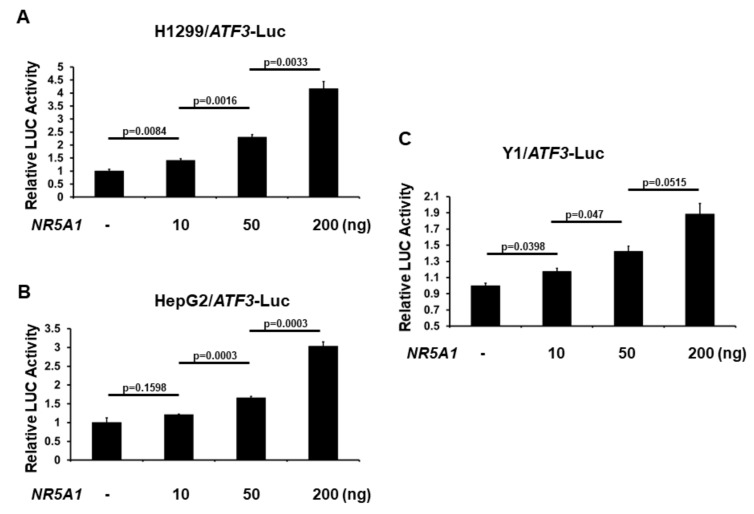
NR5A1 activates *ATF3* transcription. (**A**) H1299, (**B**) HepG2, and (**C**) Y1 cells were co-transfected, where indicated, with different amount of *NR5A1* expression plasmid and *ATF3* promoter-LUC reporter plasmid. Luciferase activities were measured 48 h after transfection and normalized with Renilla activity. Relative LUC activity was calculated and plotted.

**Figure 4 ijms-21-01429-f004:**
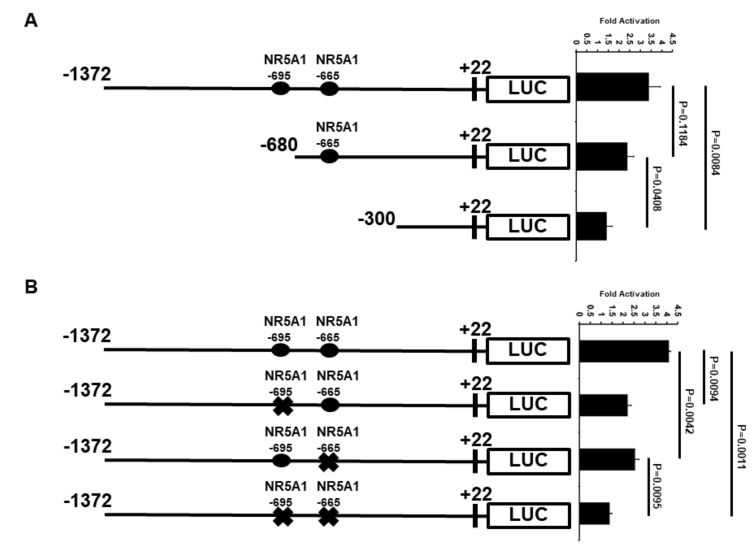
Regions of *ATF3* promoter important for transcriptional up-regulation by NR5A1. (**A**) H1299 cells were co-transfected with *ATF3* promoter deletion constructs and *NR5A1* expression plasmids. Luciferase activities were measured 48 h after transfection and normalized with Renilla activity. Relative LUC activity was calculated and plotted. (**B**) H1299 cells were co-transfected *NR5A1* expression plasmids with either −1372 wild-type, −695 RE mutated, −665 RE mutated, or both −695 and −665 REs mutated *ATF3* promoter constructs. Luciferase activities were measured 48 h after transfection and normalized with Renilla activity. Relative LUC activity was calculated and plotted.

**Figure 5 ijms-21-01429-f005:**
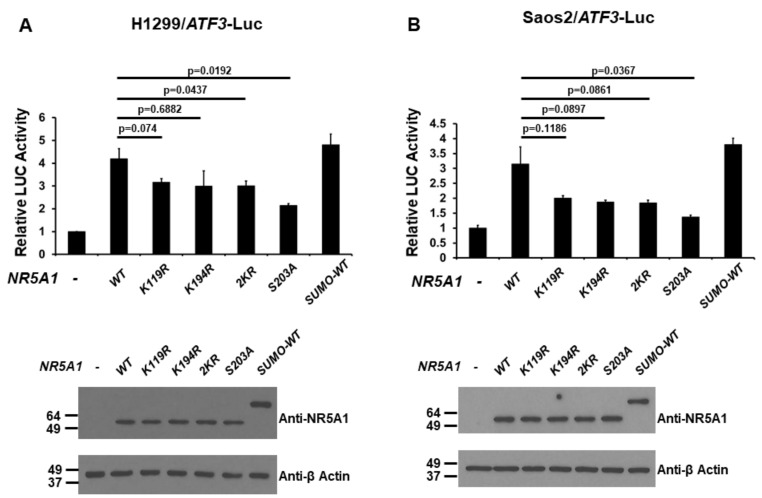
Post-translational modifications of NR5A1 regulate the activation of the *ATF3* promoter. (**A**) H1299 and (**B**) Saos2 cells were co-transfected with the *ATF3* promoter-LUC reporter plasmid and either wild-type (WT), K119R, K194R, 2KR, S203A, or small ubiquitin-related modifier (SUMO)-WT *NR5A1* expression plasmid. Luciferase activities were measured 48 h after transfection and normalized with Renilla activity. Relative LUC activity was calculated and plotted.

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
