# Peer review of "Steroidogenic Factor 1 (NR5A1) Activates ATF3 Transcriptional Activity"

_ijms, 2020, doi:10.3390/ijms21041429_

Round 1

Reviewer 1 Report

The authors report that SF1/NR5A1 activated ATF3 transcriptional activity. This is new and original; however, some experiments must be better controlled and the authors need to be more careful with some of their conclusions.

Major comments:

-SF1/NR5A1 physically interact with ATF3 and SF1/NR5A1 binds to ATF3 promoter but biologically speaking what does it mean? Does this interaction prevent SF1/NR5A1 from binding to ATF3 promoter? Is it a feed back loop? The authors never explain the relationship between these two observations.

-Figure 5: the expression of the various SF1/NR5A1 proteins need to be shown (western blot) to make sure that the differences observed are due to the mutations and not to a difference in the expression of SF1/NR5A1 wt and mutants.

-The authors claim that SF1/NR5A1 SUMO modification and phosphorylation play a role in SF1/NR5A1-mediated ATF3 expression. But they never show that the SF1/NR5A1 protein is indeed phosphorylated or sumoylated. They only have indirect proofs using mutants to prevent sumoylation and/or phosphorylation. Regarding phosphorylation, the authors could mutate the protein to mimick a permanent phosphorylation (asparctic acid). Regarding sumoylation, Ubc9 could be overexpressed to trigger sumoylation.  The experiments performed do not allow to draw any conclusion. For example, the lysine could be acetylated, instead of sumoylated and thus acetylation would be important for SF1/NR5A1-mediated ATF3 expression. In any case, the authors need to be more careful with their interpretation of the results (and writing of manuscript).

Minor comments:

-Fig3, statistical analysis needs to be performed

-The authors never explain what is a SF1/NR5A1 Reseponsive Element. How is it characterized, what is the sequence?

Author Response

Reviewer #1:

The authors report that SF1/NR5A1 activated ATF3 transcriptional activity. This is new and original; however, some experiments must be better controlled and the authors need to be more careful with some of their conclusions.

Major comments:

-SF1/NR5A1 physically interact with ATF3 and SF1/NR5A1 binds to ATF3 promoter but biologically speaking what does it mean? Does this interaction prevent SF1/NR5A1 from binding to ATF3 promoter? Is it a feed back loop? The authors never explain the relationship between these two observations.

Authors’ responses: Thanks for the reviewer’s suggestion and concerns. We discovered NR5A1 binding to ATF3 when we preformed NR5A1 binding study screening. Another AP-1 family member, JDP2, also binds to NR5A1 (data not shown), suggesting that AP-1 family may have potential binding to NR5A1. Currently our ongoing project is to dissect (1) what domain(s) of NR5A1 and ATF3 (or JDP2) bind to each other, and (2) investigate the physiological relevance of NR5A1 and ATF3 interaction. This current study was focusing how NR5A1 affects ATF3 gene expression. Currently, we do not know whether NR5A1-ATF3 interaction is associated with feed-back or feed-forward loop. It is also on our ongoing study too. Since NR5A1 is important for steroidogenesis and ATF3 has potential impact on aldosterone synthesis, our ongoing study also focuses whether NR5A1-ATF3 interaction plays a functional role for aldosterone synthesis (please see the second paragraph on discussion section).

-Figure 5: the expression of the various SF1/NR5A1 proteins need to be shown (western blot) to make sure that the differences observed are due to the mutations and not to a difference in the expression of SF1/NR5A1 wt and mutants.

Authors’ responses: Thanks for the reviewer’s suggestion and concerns. We have added western bot analysis of NR5A1 (WT and mutants) expression on Figure 5.

-The authors claim that SF1/NR5A1 SUMO modification and phosphorylation play a role in SF1/NR5A1-mediated ATF3 expression. But they never show that the SF1/NR5A1 protein is indeed phosphorylated or sumoylated. They only have indirect proofs using mutants to prevent sumoylation and/or phosphorylation. Regarding phosphorylation, the authors could mutate the protein to mimick a permanent phosphorylation (asparctic acid). Regarding sumoylation, Ubc9 could be overexpressed to trigger sumoylation.  The experiments performed do not allow to draw any conclusion. For example, the lysine could be acetylated, instead of sumoylated and thus acetylation would be important for SF1/NR5A1-mediated ATF3 expression. In any case, the authors need to be more careful with their interpretation of the results (and writing of manuscript).

Authors’ responses: Thanks for the reviewer’s suggestion and concerns. The phosphorylation and SUMOylation on NR5A1 have been extensively studied including ours. The only functional phosphorylation site on NR5A1 is S203 (Hammer et al, Mol Cell 1999). The two SUMO sites on NR5A1 are K119 and K194, with K194 the major SUMO site (Chen et al., JBC 2004; Campbell et al., MCB 2008; Ogawa et al., MBC 2009; Yang et al., MCB 2009 (my postdoc paper) and our previous publication (Wang et al., IJMS 2013). The acetylation of NR5A1 has also been studied extensively. The major acetylation sites on NR5A1 based on reports are K34, K38, and K72 (JBC 276(40): 37659-37664, 2001) and amino acids from 56-63 and 102-106 (Mol Cell Biol 25(23): 10442-10453. 2005). From published articles and our lab previous results, K119 and K194 SUMO sites on NR5A1 are not acetylation sites for NR5A1. We have used mutations mimicking phosphorylation and SUMOylation to study the role of phosphorylation and SUMOylation on NR5A1 previously. In this study, we continue to use the mutation technique to test the role of NR5A1 on ATF3 expression. Our results support the conclusion that phosphorylation at S203 on NR5A1 is a critical step for NR5A1 function with SUMOylation on a minor role. This is consistent with the previous studies including ours. Thanks.

Minor comments:

-Fig3, statistical analysis needs to be performed

Authors’ responses: Thanks for the reviewer’s suggestion and concerns. We have added statistical analysis on both Figure 3 and Figure 5.

-The authors never explain what is a SF1/NR5A1 Reseponsive Element. How is it characterized, what is the sequence?

Authors’ responses: Thanks for the reviewer’s suggestion and concerns. Using ALGGEN-PROMO website (http://alggen.lsi.upc.es/cgi-bin/promo_v3/promo/promoinit.cgi?dirDB=TF_8.3), we identified two potential NR5A1 binding sites on the human ATF3 promoter region. The two potential NR5A1 binding sites are located 695 bp (CGCCGCAGAGGTCACACCCGG) and 665 bp (TGACTTTGGACACCTTCCCC) upstream of the transcriptional start site, suggesting that NR5A1 may regulate ATF3 transcription directly. Based on this information, we truncated ATF3 promoter region and also generated NR5A1 binding site mutations for study on Figure 4. We have added this information on manuscript. Thanks.

Reviewer 2 Report

In the manuscript ''Steroidogenic factor 1 (NR5A1) activates ATF3 transcriptional activity'' the authors investigate the relation ship between the orphan receptor NR5A1 and ATF3. The authors find that NR5A1 interacts with ATF3 and may act as a cooperative activator of the ATF3 promoter via a mechanism involving Sumo and phosphorylation of NR5A1. These observations may be important in understanding the etiology of several human diseases.

The study is scientifically sound and presented in a clear and concise manner. 

One small addition that could add clarity is the inclusion of immunoblotting data showing the expression of Sumo and phospho NR5A1 mutants following transfection.

Author Response

Reviewer #2:

In the manuscript ''Steroidogenic factor 1 (NR5A1) activates ATF3 transcriptional activity'' the authors investigate the relation ship between the orphan receptor NR5A1 and ATF3. The authors find that NR5A1 interacts with ATF3 and may act as a cooperative activator of the ATF3 promoter via a mechanism involving Sumo and phosphorylation of NR5A1. These observations may be important in understanding the etiology of several human diseases.

The study is scientifically sound and presented in a clear and concise manner. 

One small addition that could add clarity is the inclusion of immunoblotting data showing the expression of Sumo and phospho NR5A1 mutants following transfection.

 Authors’ responses: Thanks for the reviewer’s suggestion and concerns. We have added western bot analysis of NR5A1 (WT and mutants) expression on Figure 5.

Round 2

Reviewer 1 Report

I would suggest that the authors be more direct in the manuscript about phosphorylation/sumoylation/acetylation experiments as their response to my comment, especially for acetylation (see quote below). It will convince the readers.

"The acetylation of NR5A1 has also been studied extensively. The major acetylation sites on NR5A1 based on reports are K34, K38, and K72 (JBC 276(40): 37659-37664, 2001) and amino acids from 56-63 and 102-106 (Mol Cell Biol 25(23): 10442-10453. 2005)."

Author Response

I would suggest that the authors be more direct in the manuscript about phosphorylation/sumoylation/acetylation experiments as their response to my comment, especially for acetylation (see quote below). It will convince the readers.

"The acetylation of NR5A1 has also been studied extensively. The major acetylation sites on NR5A1 based on reports are K34, K38, and K72 (JBC 276(40): 37659-37664, 2001) and amino acids from 56-63 and 102-106 (Mol Cell Biol 25(23): 10442-10453. 2005)."

Authors’ responses: Thanks for the reviewer’s suggestion and concerns. We have added this information on the 4th paragraph of the discussion section to highlight the acetylation of NR5A1. Thanks.